# Novel Theranostic Approaches Targeting CCR4-Receptor, Current Status and Translational Prospectives: A Systematic Review

**DOI:** 10.3390/ph16020313

**Published:** 2023-02-17

**Authors:** Joana Gorica, Maria Silvia De Feo, Ferdinando Corica, Marko Magdi Abdou Sidrak, Miriam Conte, Luca Filippi, Orazio Schillaci, Giuseppe De Vincentis, Viviana Frantellizzi

**Affiliations:** 1Department of Radiological Sciences, Oncology and Anatomo-Pathology, Sapienza, University of Rome, 00161 Rome, Italy; 2Department of Nuclear Medicine, Santa Maria Goretti Hospital, 04100 Latina, Italy; 3Department of Biomedicine and Prevention, University Tor Vergata, 00133 Rome, Italy

**Keywords:** CCR4, theranostics, CXCR4, chemokine receptor, immunotherapy

## Abstract

Background: With the high mortality rate of malignant tumors, there is a need to find novel theranostic approaches to provide an early diagnosis and targeted therapy. The chemokine receptor 4 (CCR4) is highly expressed in various tumors and plays an important role in tumor pathogenesis. This systematic review aims to provide a complete overview on clinical and preclinical applications of the CCR4 receptor as a target for theranostics, using a systematic approach to classify and assemble published studies performed on humans and animals, sorted by field of application and specific tumor. Methods: A systematic literature search of articles suiting the inclusion criteria was conducted on Pubmed, Scopus, Central, and Web of Science databases, including papers published from January 2006 to November 2022. Eligible studies had to be performed on humans and/or in vivo/in vitro studying CCR4 expression in tumors. The methodological quality was assessed through the Critical Appraisal Skills Programme (CASP) assessing only the studies performed on humans. Results: A total of 17 articles were screened. The articles were assessed for eligibility with the exclusion of 4 articles. Ultimately, 13 articles were selected for the qualitative analysis, and six articles were selected for the critical appraisal skills program. Conclusions: The development of new radionuclides and radiopharmaceuticals targeting CCR4 show promising results in the theranostics of CCR4 sensible tumors. Although to widen its use in clinical practice, further translation of preclinical to clinical data is needed.

## 1. Introduction

The clinical need to meet the increasing number of oncologic patients and the high mortality rate of such tumors have led to vast research in finding new molecular targets for early diagnosis and therapy, combining both in a unique approach, namely “theranostics” or “theragnostics” [1,2]. Since mortality and the prevalence of malignant tumors have grown over time, conventional diagnostic techniques such as routine imaging, histopathological, and serological tests have become less effective. Additionally, due to limitations in efficacy and substantial side effects, conventional treatments for malignant tumors such as surgery, chemotherapy, and radiotherapy are unable to meet the urgent need for effective care. Molecular imaging uses targeted molecular imaging probes to obtain qualitative and quantitative in vivo imaging of the tumoral tissues, to measure the presence and the development of the disease [3,4]. The idea of employing radionuclide-labeled peptide-binding receptors has generated a lot of research in nuclear medicine. Peptide has a low molecular weight compared to proteins, and antibody fragments, showing the ability to rapidly differ in target tissues, and can generate a high tumor-to-background ratio [5]. Cell migration and homing are controlled by chemokines and their receptors in the body. At least 44 chemokines have been identified in humans, and these chemokines have been divided into four subfamilies based on the conserved cysteine motifs found at the N-terminus: CXC, CC, C, and CX3C. The receptor for two CC chemokine ligands (CCLs)—CCL17 (also known as thymus- and activation-regulated chemokine) and CCL22—is the CC chemokine receptor 4 (CCR4) (macrophage-derived chemokine) [6]. A very promising molecule is the chemokine receptor 4 (CCR4). It is overexpressed in many types of cancers, including hematopoietic, epithelial, mesenchymal, breast, thyroid, etc. CCR4 is part of the G protein-coupled receptor (GPCR) and is a member of the 7-transmembrane domain family of receptors. CCR4 has been used as a target for many drugs of immunotherapy. It is present on the surface of the cell, and it consists of 352 amino acid residues, containing an N-terminal, C-terminal, and 7-transmembrane domains, and three extracellular and intracellular loops. CCR4 passes the signal within the cell by combining with the signaling chemokine, this way it controls cellular growth and the immune system [3]. The aim of this systematic review is to show the vast utility and application of this molecule in different types of tumors, through research from pre-clinical to clinical data.

## 2. Materials and Methods

### 2.1. Search Strategy and Study Selection

This systematic review was designed following PRISMA guidelines. An online literature search of articles that suited the inclusion criteria was conducted on Central (Cochrane Library), Pubmed, Web of Science, and Scopus databases and included papers published from January 2006 to November 2022. The following search keywords were applied for each database: “CCR4 receptor theranostics” or “CXCR4 chemokine receptor target” or “CCR4 target theranostics”. Eligible studies had to be performed on humans and/or in vitro/in vivo, and for inclusion in the qualitative analysis English language was mandatory. The references of the provided articles were also examined in order to find out any additional relevant studies.

### 2.2. Data Extraction and Methodological Quality Assessment

General data about the article, specifically authors, journal, year of publication, country, study design, and patient characteristics, were retrieved for all the included studies. The methodological quality was assessed through the Critical Appraisal Skills Programme (CASP) tool, widely used and recommended for systematic reviews of diagnostic accuracy by the Agency for Healthcare Research and Quality, Cochrane Collaboration (Cochrane Handbook for Systematic Reviews of Diagnostic Test Accuracy), and the U.K. National Institute for Health and Clinical Excellence.

## 3. Results

### 3.1. Analysis of the Evidence

The resulting PRISMA search strategy is shown in Figure 1.

From the systematic literature search, a total of 17 articles were assessed for eligibility by examining each abstract. One article was excluded because it was a duplicate in two different search sites and three articles were excluded due to the lack of full text in English. Finally, 13 papers were selected for the systematic review. Only six papers were suitable for CASP analysis, for an overall number of 185 CCR4-positive oncologic patients. From the analysis of the selected papers, three main thematic areas were identified, diagnosis, radiation, and therapy. The findings of the selected papers for each thematic area are described in the following paragraphs.

### 3.2. Study Characteristics and Methodological Quality Assessment

The 13 selected papers were published from 2006, the year of publication of the first study regarding CCR4 with ^177^Lu-pentixather, to 2022, the clinical trials with Mogamulizumab. Only six studies were conducted in humans, and the number of enrolled patients ranged from 1 to 64. Three studies were conducted in vivo, using mice and six studies were conducted in vitro. The Critical Appraisal Skills Programme (CASP) was used in human studies. Considering the methodological quality of the studies included the following. four of six studies satisfied at least nine and only one of six of the twelve CASP domains for the bias risk assessment (Table 1). Analyzing the results within each bias assessment domain, five studies obtained a low concern of bias and one study showed high risk in some of the domains. Taking into consideration all bias assessment domains, only three studies reported more than two unclear results, in relation to insufficient information given to achieve an optimal methodological protocol appraisal. In conclusion, the risk for global applicability was mainly low. The studies’ characteristics are described in Table 2.

## 4. Discussion

### 4.1. CCR4

CCR4 also known as C-C chemokine receptor 4, produces a protein that is a member of the G-protein-coupled receptor family. It is a CC chemokine receptor for MIP-1, RANTES, TARC, and MCP-1. Chemokines are a family of tiny, structurally similar polypeptide molecules that control how different kinds of leukocytes move around inside their cells. The chemokines also have important impacts on cells of the central nervous system and endothelial cells engaged in angiogenesis or angiostasis, as well as on immune system development, homeostasis, and function [19]. CCR4 is part of the G protein-coupled receptor (GPCR) and is a member of the 7-transmembrane domain family of receptors. CCR4 has been used as a target for many drugs of immunotherapy. It is present on the surface of the cell, and it consists of 352 amino acid residues, containing an N-terminal, C-terminal, and 7-transmembrane domains, and three extracellular and intracellular loops. CCR4 passes the signal within the cell by combining with the signaling chemokine; this way it controls cellular growth and the immune system. CCR4 overexpression is associated with many types of tumors. The CCR4 interaction with CCL12 activates intracellular signaling that activates various pathways such as NF-Kb, RAS-MAPK, JAK-STAT, P13K-AKT-mTOR, and PLC-Ca^2+^, which can regulate angiogenesis, tumor cell proliferation, and apoptosis inhibition. The activation of mitogen-activated protein kinase (MAPK) activates the c-myc transcription factor, which regulates CCR4 overexpression, therefore leading to positive and negative feedback, promoting tumor growth and cell proliferation. Cell proliferation is also generated through the activation of P13K-AKT and RAF-MEK-ERK pathways. Apoptosis is inhibited through the activation of NF-kB, Wnt pathways, and the upregulation of Bcl-2 proteins. MAPK, ERK1/2, and P13K activate the secretion of MMP which promotes the ability of tumors to metastasize. Due to the crucial role of CCR4 in tumors, various target inhibitors have been studied [3,10]. See Figure 2.

#### 4.1.1. CCR4 Tissue Expression

Inflammatory responses cause an upregulation of chemokine receptors during immune activation, and the expression of homing receptors is essential for the movement of immune cells. To facilitate their migration to lymphoid organs for antigen interactions and to exert local immune responses, T cells express certain chemokine and homing receptors. Chemokine receptors on T cells differ in expression, reflecting the specific T cell subset that they belong to [20]. CCR4 participates in T cell lung imprinting and is connected to Th2 differentiation. Spoerl et al., discovered that individuals with severe COVID-19 infection, primarily characterized by pulmonary failure, had significantly higher levels of the lung-homing receptor CCR4 and the proinflammatory receptor CCR5 on CD8 T cells [20]. CCR4 is expressed on non-cancerous CD45RA/Foxp3high/CD4 eTregs. Since eTreg-mediated immune evasion mechanisms appear to be crucial in the development of several types of malignancies, CCR4 is a desirable target for cancer immunotherapy. Patients with squamous cell carcinoma of the lung have higher levels of CCR4 expression in these cells, which is a predictor of survival and relapse [21]. Lei Wang et al., discovered CCR4 as a possible predictive biomarker for survival and recurrence of patients with pN0 oral tongue cancer. Therefore, CCR4 may be a potential therapeutic target for those with early-stage cancer [22]. Cheng X et al. demonstrated that the upregulation of chemokine receptor CCR4 is associated with Human Hepatocellular Carcinoma malignant behavior [23]. Ou B et al. demonstrated that CCR4 promotes metastasis via the ERK/NF-κB/MMP13 pathway and acts downstream of TNF-α in colorectal cancer [24]. Furthermore, the expression of chemokine receptor 4 was associated with a bad prognosis in renal cell carcinoma in the study of Liu Q et al. [25]. It is possible that tumor-induced immunosuppression is influenced by the abnormal expression of CCR4 in human gastric cancer. Feasibly suppressing CCR4 expression could be a successful treatment for human stomach cancer [26]. In both animal models and human samples, the crucial part played by the CCR4 axis in immunological suppression by Treg cells has been thoroughly described [27,28]. For instance, tolerance induction by anti-CD154 with donor-specific transfusion, which was followed by the intragraft expansion of Foxp3+ T cells, was not accomplished in Ccr4/recipients in an allogeneic heart transplantation paradigm [29]. In a rat adoptive transfer model of inflammatory bowel illness, Ccr4/Treg cells failed to aggregate in the mesenteric lymph nodes to suppress colitis [30]. In Ccr4/animals, Treg cell accumulation in the skin and airways of the lungs was restricted, leading to severe inflammatory disorders [28]. CCR4 is the only receptor for the chemokines CCL22/Monocyte Derived Chemokine (MDC) and CCL17/Thymus- and Activation-Regulated Chemokine (TARC). Naive T-lymphocytes show little to no CCR4, as CCR4 is largely expressed on CD4+CD8+ thymocytes and appears to be downregulated upon passage from the thymus. Since CCL22 and CCL17 are both expressed in the thymus, one function for this particular set of chemokines and receptors may be to control the migration of CCR4+ thymocytes inside the thymus during the education and differentiation of T-lymphocytes. Additionally, it has been demonstrated that primary leukemia cells secrete CCL22 and CCL17 in response to CD40 ligation, indicating that the chemokines may direct B-cell responses that are dependent on T lymphocytes in secondary lymphoid tissues by enlisting helper T lymphocytes and dendritic cells. Along with its ligands, CCR4 has been associated to the pathophysiology of allergic skin illnesses including atopic dermatitis (AD), where elevated blood levels of CCL17 are linked to disease activity. CCL17 is a marker for cells that express the cutaneous lymphocyte-associated antigen. Additionally, skin biopsies show that CCR4+ cells are invading both the dermis and the epidermis in this illness [31].

#### 4.1.2. CCR4 in Animal Studies

CCR4 is regarded as a marker of Th2 cells because it is the receptor for the chemokines CCL17 (thymus and activation-regulated chemokine, TARC) and CCL22 (macrophage-derived chemokine, MDC). Eosinophils and skin specific T cells are the main cell types that express CCR3 and CCR8, however they are also linked to Th2 cells. According to numerous findings, CCR4 and its ligands play a significant role in MS. Additionally, Th17 cells, Tregs, macrophages, and dendritic cells (DCs) express the CCR4 receptor. CCR4 and its ligands may be pathogenic, especially when it comes to DCs and Th17 cells. In comparison to controls, MS patients had greater CSF levels of CCL17 but lower serum levels. Despite the fact that the levels of CCL22 in the CSF were comparable across male and female MS patients and controls, higher amounts of CCL22 were found in the CSF of female MS patients. CCR4 and CCL22 are expressed more frequently in the CNS in EAE mice. While wild-type C57BL/6 mice developed severe clinical indications of MOG-induced EAE, CCR4-deficient (CCR4/) C57BL/6 mice developed resistance and less severe neuropathology, as well as lower levels of CD4+ T cells and macrophages in the CNS. Additionally, prophylactic administration of CCR4 antagonist to C57BL/6 mice improved MOG-induced EAE with a decline in neuropathology. Passive MOG-specific T cell transfer resulted in recipient wild-type C57BL/6 mice developing EAE with a similar severity and frequency to those that received encephalitogenic T cells from wild-type mice. On the other hand, the intracerebral injection of CCR4-bearing DCs, but not CCR4/DCs, aggravated EAE in MOG-sensitized CCR4/animals, which developed minimal clinical scores [32].

### 4.2. CCR4-Targeted Molecular Imaging

Molecular imaging has become crucial in the medical field, especially in the diagnosis of malignant diseases. The goal of molecular imaging is to monitor in real-time biochemical and biological processes taking place in the organism. Through targeted molecular imaging probes, we can obtain qualitative, quantitative, and dynamic in vivo imaging, to reveal the pathogenesis and the status of the disease [3]. It also includes multiple image-capture techniques in combination with data retrieved from other medical and nonmedical sources. Single photon emission tomography (SPECT) and positron emission tomography (PET) are the main imaging technologies used. Radionuclide-labeled probes are used to identify and monitor tumors and monitor therapeutic response. The use of highly targeted therapies reduces the need for additional therapies, such as chemotherapy and radiotherapy, and the side effects that they can cause.

#### 4.2.1. SPECT/CT CCR4-Target Imaging

Single photon emission computed tomography/computed tomography (SPECT/CT) combines two different diagnostic scans into one for a more accurate view of the body region being examined and to track the metabolic processes of the body. The CT scan provides a better anatomical resolution. Single photon emission computed tomography has cost-effective advantages and high sensitivity in the ability to track physiopathological processes in vivo. There are a variety of radiopharmaceuticals and isotopes that can be used in SPECT/CT, some of which have been shown to have utility in tracking CCR4 sensible tumors. Hanaoka et al. developed a radiopharmaceutical for the imaging of CCR4 sensible tumors. They developed a precursor for radiolabeled peptides, containing 14-residue peptidic CXCR4 inhibitor, Ac-TZ14011 creating the radiopharmaceutical ^111^In-DTPA-Ac-TZ14011. Imaging showed a major intra-tumor biodistribution of the tracer compared to other healthy tissue and blood [15]. Kuil et al., developed a hybrid peptide dendrimer, containing Ac-TZ14011, Cy5.5-like fluorophore, and a DTPA chelate which reduced a specific muscle uptake in molecular imaging [16]. Lesniak et al. evaluated another imaging agent that targets CCR4, POL3026 peptidomimetic template. Through single photon emission computed tomography (SPECT/CT) imaging, biodistribution, and in silico and in vitro studies show [^111^In]POL-D and [^111^In]POL-PD significant uptake in U87-stb-CXCR4 tumors compared to the control U87 tumors at 90 min and 24 h post-injection [17]. Other SPECT/CT isotopes targeting CCR4 with promising results are ^99m^Tc-HYNIC, a co-ligand with bifunctional chelator, hydrazino nicotinic acid [33] and ^67^GA-DOTA-TZ2 [34].

#### 4.2.2. PET CCR4-Target Imaging

Positron emission tomography is broadly employed as a molecular imaging technique, using radiopharmaceuticals as tracers to track the metabolism of whole tissues and focal lesions to diagnose diseases. PET scans are usually combined with computed tomography (PET/CT) and magnetic resonance imaging (PET/MRI). Through the fusion with anatomical structure imaging of CT or MRI, we can obtain information regarding function and metabolism while having high spatial resolution. There is a high variety of PET radiotracers. Lutetium-177 is a low-energy B-emitting radiotracer with a half-life of 6 or 7 days. It has the ability to bind with DOTA/NOTA chelates making it ideal for radiotracer therapy. ^177^Lu-DOTA-TATE and ^177^Lu-PSMA-617 can be targeted to a peptide receptor and are ideal as they have a high affinity for tissues expressing such peptides [18]. ^68^Ga-pentixafor is a promising positron emission tomography (PET) agent targeting CCR4. Pentixafor has a similar structure to CCR4, providing binding stability to plasma proteins. Herrman et al. used ^177^Lu and ^90^Y bound with pentixafor to cure Multiple Myeloma (MM) in three patients with advanced disease. The results were promising with one full recovery and a partial one [12]. Other radionuclides have a high affinity for CCR4 sensible tumors and have bifunctional chelating groups; ^18^F-T140, ^18^F-AIF-NOTA-T140, ^64^Cu-T140-2D, ^64^Cu-DOTA-NFB, ^64^Cu-NOTA-NFB, ^68^Ga-DOTA-CPCR4-2 and ^68^Ga-NOTA-NFB [3].

### 4.3. Radiation-Enhanced Expression of Chemokine Receptor CCR4

Radiation therapy, also called radiotherapy, is a treatment that uses high doses of radiation to kill cancer cells and reduce the size of tumors. Radiotherapy is commonly used for the treatment of many types of tumors, by itself or in addition to chemotherapy. Its success depends on accurate and reproducible dose delivery to the target volume and minimization of concomitant doses to healthy tissues. There is, therefore, a requirement for robust dosimetry for all parts of the body, and for all treatment modalities, for patients undergoing radiotherapy. Radiotherapy uses ionizing radiation to eliminate cancer cells through apoptosis, necrosis, and immunogenic cell death. Radiotherapy causes alterations in gene expression in tumor cells, and it can involve the immune system [7]. Irradiated tumor tissues and cancer cells recruit chemokines such as CCL22 and CXCL12 and produce a range of cytokines such as TNF-α, TGF-β, IL-1, IL-6, and GM-CSF which could recruit tumor-infiltrating lymphocytes (TILs) into radiation site. Radiation produces pro-inflammatory factors to recruit TILs and change the inflammatory microenvironment. Tissue radiation generates ROS and kinase activation associated with DNA damage, resulting in the expression of NK cell-activating ligands (NKALs) that activate and secrete the chemokines XCL1, FLT3LG, and CCL5 to recruit dendritic cells (DCs), which, induces tumor suppression in the tumor microenvironment [7]. C-C class chemokine 22 (CCL22) is macrophage-synthetized; its receptor is C-C chemokine receptor 4 (CCR4), and it is expressed on the surface of Th2 cells and regulatory T cells (Treg). High levels of CCL22 have been found in various types of human tumors such as breast, lung, gastrointestinal, and nasopharyngeal tumors, B–CLL, and Hodgkin lymphoma. Radiation treatment increases intracellular and extracellular expression of CCL22, these factors indicate that radiotherapy has immune-stimulating properties. Li H et al. studied the prognosis of patients expressing both CCL22 and CCR4 through Kaplan—Meier survival analysis. The study revealed that such patients had a better prognosis, which was not expected due to Treg cell recruitment by CCL22 and CCR4 [7].

### 4.4. Mogalizumab, Humanized Anti-CCR4 Monoclonal Antibody, and Its Applications

#### 4.4.1. Adult T Cell Leukemia and Sézary Syndrome

Adult T cell Leukemia (ATL) has a very poor prognosis. CCR4 is present in the neoplastic cells of most patients with this type of tumor. Mogamulizumab (see Figure 3) is a novel monoclonal antibody used in immunotherapy targeting the humanized CCR4. This antibody enhances antibody-dependent cellular toxicity through its defucosylated Fc region. Mogamulizumab mediates antibody-dependent cellular toxicity, but it does not mediate complement-dependent cytotoxicity or other antitumor activities. The main effector cells of Mogalizumab are considered to be the natural killer (NK) cells [9]. Mogamulizumab has been proposed as a first-line therapy in ATL patients not suitable for allogeneic stem cell transplantation and patients with refractory ATL. Mogamulizumab has been approved by the Food and Drug Administration and the European Medicines Agency for the treatment of patients with Sézary Syndrome (SS) in 2018 [9]. SS and ATL, both T cell tumors, share similar genetic alterations. Tanaka et al. studied 64 ATL patients treated with Mogalizumab. Forty-five patients received combined therapy with Mogalizumab and vincristine, cyclophosphamide, doxorubicin, prednisone; doxorubicin, ranimustine, prednisone; vindesine, etoposide, carboplatin, prednisone or with cyclophosphamide, doxorubicin, vincristine, prednisolone, whereas 19 received Mogamulizumab monotherapy. Nine patients received an allogeneic hematopoietic stem cell transplant after Mogamulizumab treatment. The patients were evaluated for gene alterations and clinical responses correlated to specific gene alterations. CCR4 was among the 11 genes altered in ATL patients and also one of the six genes altered in SS. CCR4 alterations were present in 25–30% of ATL patients, in addition, this alteration was also correlated to a higher clinical response rate. Most likely, the alterations in the C-terminus guide to a weakened CCR4 internalization upon ligand binding, leading to a higher surface expression of the ligand and leading to increased availability of target molecules for Mogalizumab [9].

#### 4.4.2. Cutaneous Lymphomas

A class of non-Hodgkin lymphomas is primary cutaneous lymphomas, such as mycosis fungoides (MF) and Sézary syndrome (SS). Mycosis fungoides is one of the most common types of Cutaneous T Cell Lymphomas (CTCL). MF originates in the peripheral epidermotropic T cells, in specific in the memory T cells (CD45RO+), expressing the T cell receptor (TCR) and CD4+ immunophenotype [35]. Vaidya et al. in their 2022 publication showed that MF has an incidence of six cases per million per year in the United States and Europe. That would be accountable for 4% of all cases of non-Hodgkin lymphomas. It is more common in male adults over 50 years of age. The pathology is more common amongst dark skinned races rather than Caucasians or Asians [36]. Sezary syndrome is accountable for 3% of all T cell cutaneous lymphomas and it expresses three specific findings: lymphadenopathy, erythroderma with pruritus, and most specifically, atypical circulating lymphocytes called Sezary or Lutzner cells. Sezary Syndrome is part of the leukemic phase of T cell cutaneous lymphomas that rarely compromises the bone marrow; its compromission is found only in advanced forms of the disease. In the World Health Organization-European Organization for Research and Treatment of Cancer (WHO-EORTC) classification, MF and SS are listed as separate diseases [37]. Monochemotherapy with gemcitabine or pegylated liposomal doxorubicin is effective, but not in all patients, and the duration of response is limited. Allogeneic hematopoietic stem cell transplant is resolutive, but it has a high treatment-associated mortality rate and relapses are not uncommon. Therefore, new targeted therapies have been developed targeting surface molecules expressed on the surface of tumor cells, such as CCR4, HDACs, CD30, and CD25, CD52, specific for MF and SS. Mogamulizumab, whose antitumor activity is mediated by antibody-dependent cellular cytotoxicity, is a humanized monoclonal antibody against CCR4. Mogamulizumab has been approved for CCR4+ ATL and for peripheral T cell lymphoma (PTCL) and cutaneous T cell lymphoma (CTCL). Sugaya et al. enrolled seven patients with MF, and their overall response (ORR) was 28.6% with no complete response (CR) [8]. Duvic et al. conducted a phase ½ study in 41 pre-treated patients with CTCL. Thirty-eight patients were treated with Mogalizumab, the ORR was 36.8%: 47.1% in SS (n = 17), and 28.6% in MF (n = 21). Ogura et al. conducted a multicenter phase II study of Mogalizumab. They enrolled 38 patients, 37 received Mogalizumab therapy and exhibited clinically meaningful outcomes in patients with CTCL and PTCL [11]. The MAVORIC (Mogamulizumab anti-CCR4 Antibody Versus ComparatOR In CTCL) study is a phase 3 multicentric clinical trial for relapsed or refractory MF-SS. The study revealed that Mogamulizumab has a better effect on survival than the HDAC inhibitor vorinostat. Response rates based on tissues were: 68% in blood, 17% in lymph nodes 0.42% in the skin, and 0% in viscera. The higher response rate was seen in the blood, due to the presence of natural killer cells, neutrophils, or monocytes, which are essential for ADCC. Common side effects were fever, pruritus, nausea, skin eruptions, infusion reactions, and lymphopenia. In 2018, Mogamulizumab has been approved for the treatment of patients with CTCL who have received at least one prior systemic therapy by the FDA and the European Medicines Agency [8].

## 5. Future Perspectives

The chemokine receptor CCR4 is expressed on various cancer cells and its expression is correlated with the progression of the disease and prognosis. When it binds to its ligand CCL12, it activates different cancer development pathways and has a crucial role in the relapse of the tumor after chemotherapy. Particularly, in acute myeloid leukemia (AML), high levels of CCR4 are connected with high relapse and low survival. Consequently, the inhibition of CXCR4 by siRNA silencing and CXCR4 antagonists are promising strategies in AML therapy. New therapeutic strategies targeting CCR4 include genome editing to knock out CCR4 or the use of short artificial, single-strand oligonucleotide sequences capable of binding to biological molecules, namely aptamers [38,39]. Xia-He Ren et al. developed short palindromic repeats-CRISPR-associated protein 9 (CRISPR-Cas9) systems for genome editing in cancer cells. A multifunctional leukemia-targeting gene vector, Herein, was developed to deliver the CRISPR-Cas9 in leukemia cells, by using biomacromolecules with ideal biocompatibility. The CRISPR-Cas9 plasmid was improved with protamine, and mucin-1 (MUC1) aptamer incorporated alginate and T22-NLS peptide. T22 has a binding affinity to CCR4. Herein can induce self-assembly via the electrostatic interaction and, in such a way, it introduces cancer-targeting aptamer and peptide into the delivery system. It is a dual delivery system as it targets CCR4 overexpressed leukemia cells and MUC1 providing CCR4 knockout [38]. Genome editing plasmids seem to be the future of personalized therapy.

## 6. Conclusions

CCR4 is an important target for the diagnosis and therapy of various tumors. Radiopharmaceuticals that target such chemokines have an important role in identifying patients that have CCR4-positive tumors and that can respond to CCR4-targeted therapy. Theranostic radionuclides such as ^68^Ga-pentixafor and ^177^Lu-PSMA/DOTA/NOTA and humanized monoclonal antibodies like Mogamulizumab show promising results for various tumors, especially in PTCL and CTCL. The development of new radionuclides and radiopharmaceuticals targeting CCR4 show promising results in the theranostics of CCR4 sensible tumors. Although, to widen its use in clinical practice, further translation of preclinical to clinical data is needed.

## Figures and Tables

**Figure 1 pharmaceuticals-16-00313-f001:**
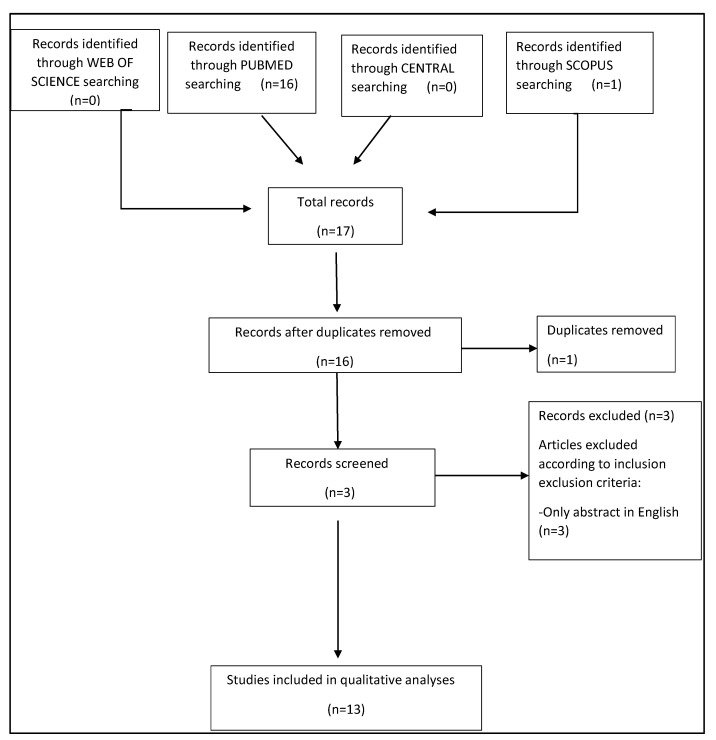
PRISMA flow-chart.

**Figure 2 pharmaceuticals-16-00313-f002:**
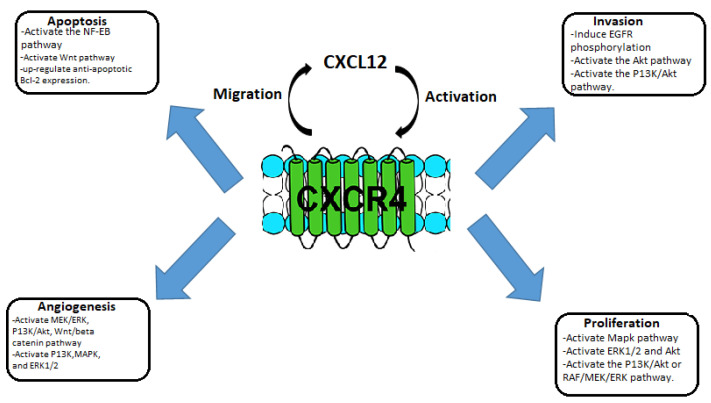
Schematic representation of the role of CXCL12/CCR4 in tumorigenesis and metastasis.

**Figure 3 pharmaceuticals-16-00313-f003:**
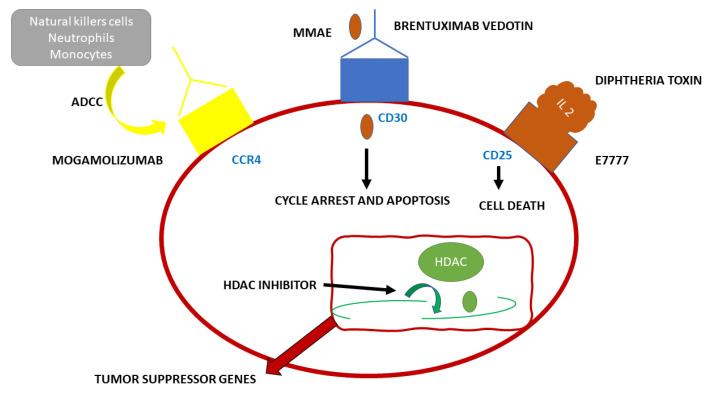
Schematic representation of the functioning of the monoclonal antibody Mogamulizumab.

**Table 1 pharmaceuticals-16-00313-t001:** CASP analysis.

	1. Was There a Clear Question for the Study to Address?	2. Was There a Comparison with an Appropriate Reference Standard?	3. Did All Patients Get the Diagnostic Test and Reference Standard?	4. Could the Results of the Test Have Been Influenced by the Results of the Reference Standard?	5. Is the Disease Status of the Tested Population Clearly Described?	6. Were the Methods for Performing the Test Described in Sufficient Detail?	7. What Are the Results?	8. How Sure Are We about the Results? Consequences and Cost of Alternatives Performed?	9. Can the Results Be Applied to Your Patients/the Population of Interest?	10. Can the Test be Applied to Your Patient or Population of Interest?	11. Were All Outcomes Important to the Individual or Population Considered?	12. What Would Be the Impact of Using This Test on Your Patients/Population?
Li H et al., 2020 [7]	☺	?	?	?	☺	☺	☺	☺	☹	☹	☺	It may have a positive outcome in therapeutic accuracy and prognosis in human patients
Sugaya et al., 2021 [8]	☺	☺	☺	?	☺	☺	☺	?	☺	☺	☺	It may lead to a better-targeted therapy
Tanaka et al., 2022 [9]	☺	☺	☺	☺	☺	☺	☺	☺	☺	☺	☺	Immunotherapy with a more specific targeted receptor and better prognosis for the patient
Ogura et al., 2014 [10]	☺	?	☺	?	☺	☺	☺	☺	☺	?	☺	Immunotherapy response rate monitorization
Duvic et al., 2015 [11]	☺	?	☺	☺	☺	☺	☺	?	☺	?	☺	Utility in targeted immunotherapy
Herrmann et al., 2016 [12]	☺	☺	☺	?	☺	☺	☺	☺	☺	☺	☺	Promising results

☺ Low risk; ? Unknown; ☹ High risk.

**Table 2 pharmaceuticals-16-00313-t002:** Studies’ characteristics.

Authors	Year	Type of Article	Nr of Pts	Main Cancer Tissue Expression	Main Molecule	Main Finding
Zhang et al. [13]	2021	Original article	In vitro	Pituitary adenoma	mTORC2/ CCL17	The CCL17/CCR4/mTOCR1 axis may serve as a potential therapeutic target for pituitary adenoma.
Sugaya et al. [8]	2021	Review	38	Cutaneous T cell lymphoma	CD158k, JAK, PIK3, target of rapamycin, and microRNAs	Personalized therapy based on the detection of the genetic signatures of tumors and inhibition of the most suitable target molecules constitutes a future treatment strategy for MF/SS.
Li et al. [7]	2020	Original article	1	Nasopharyngeal Carcinoma	CCL22	The radiation-enhanced release of CCL22 from NPC cells promotes migration of CCR4 + effector CD8 T cells, which might partially be associated with radiation therapy-mediated antitumor immunity.
Tanaka et al. [9]	2022	Original article	64	Lymphoma, T Cell, Peripheral and Cutaneous	Mogamulizumab	Patients with CCR4 alterations or without CCR7 alterations exhibited a more favorable clinical response to Mogamulizumab.
Hua et al. [14]	2014	Research	In vivo	T cell acute lymphoblastic leukemia (T-ALL)	Forkhead box O_3_/mTORC2	The inactivation of mTORC2 causes the overexpression of forkhead box O_3_ and its downstream effectors and eases the progression of leukemia in T-ALL mice.
Liu et al. [3]	2019	Review	In vitro	CCR4 expressing tumors	CXCR4 peptides	The potential of CCR4 as a theranostic agent.
Hanaoka et al. [15]	2006	Research	In vitro	Pancreatic cancer	^111^In-DTPA-Ac-TZ14011	^111^In-DTPA-Ac-TZ14011 would be a potential agent for the imaging of CXCR4 expression in metastatic tumors in vivo
Kuil et al. [16]	2011	Research	In vitro	CCR4 expressing tumors	Hybrid peptide dendrimers/ cyclic Ac-TZ14011 peptide	Biodistribution studies revealed that the additional peptides in the dimer and tetramer reduced nonspecific muscle uptake.
Lesniak et al. [17]	2015	Research	In vivo	CXCR4 expressing subcutaneous U87 tumors	[^111^In]POL-D and [^111^In]POL-PD	POL3026 is a promising template to develop new imaging agents that target CXCR4.
Schottelius et al. [18]	2017	Research	In vivo	Multiple Myeloma	[^68^Ga]pentixafor/ [^177^Lu]pentixather	High clinical potential for [^68^Ga]pentixafor/[^177^Lu]pentixather in CCR4 expressing tumors.
Herrmann et al. [12]	2016	Research	3	Multiple Myeloma	^177^Lu-pentixather/ ^90^Y-pentixather	CXCR4-targeted radiotherapy with pentixather appears to be a promising novel treatment option in combination with cytotoxic chemotherapy and autologous stem cell transplantation, especially for patients with advanced multiple myeloma.
Ogura et al. [10]	2014	Clinical Trial, Phase II	38	Lymphoma, T Cell, Peripheral and Cutaneous	Mogamulizumab	Mogamulizumab showed meaningful antitumor activity in patients with relapsed PTCL and CTCL.
Duvic et al. [11]	2015	Clinical Trial, Phase II	41	Cutaneous T cell lymphoma	Mogamulizumab	No dose-limiting toxicity was observed with the use of Mogamulizumab.

## Data Availability

Not applicable.

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
