# Peer review of "Novel Theranostic Approaches Targeting CCR4-Receptor, Current Status and Translational Prospectives: A Systematic Review"

_pharmaceuticals, 2023, doi:10.3390/ph16020313_

Round 1
Reviewer 1 Report
Dear authors,
in my opinion a review that collects information for a topic like tharanostic related to CCR4-receptor is really interesting. I would like to inform me why the number of papers that you used is just 22. I think that there are many other papers after searhing in different databases using these keywords.
Moreover, you should present with more details the reasons that some research studies are characterized as biased.
Finally, I suggest to present your opinion in discussion part, as well, in order to critisize the presented results. Your discussion is just a presentation of pubished results. In my opinion, you either change the title of this part or add some points with your opinion related to the topic. It may improve your article.
Kind regards,
Author Response
First of all, I want to thank you for the attention shown and the precious suggestions, which we welcome, which certainly improve the quality and comprehensibility of our work.
All suggested corrections have been made in the manuscript.
Reviewer 2 Report
The systematic review summarized the clinical and preclinical applications of CCR4-receptor as a target for theranostics. The content is advanced, arrangement is reasonable. It will be a good help for readers to refer to. The title of Table 1 should be added.
Author Response
I want to thank you for the attention shown and the precious suggestions, which we welcome, which certainly improve the quality and comprehensibility of our work.
All suggested corrections have been made in the manuscript
Reviewer 3 Report
The review written “Novel theranostic approaches targeting CCR4-receptor, current status, and translational prospectives: a systematic review”. Below are a series of comments from the authors. Here are my observations:
1. Authors should include a table/cartoon showing the pathways targeted by the CCR4 receptor. Diagrammatic representations are very important in a review. You should have at least two diagrammatic representations.
2. The introduction is very short; authors need to elaborate on each section, and authors should provide additional references.
3. Authors should include the latest references of CCR4 expression on various cancer cells.
4. Explain the role of CCR4 expression in experimental animal models.
5. Authors should add recent references to CCR4-receptor signaling
6. I suggest the authors revise the conclusion portion.
Author Response
I want to thank you for the attention shown and the precious suggestions, which we welcome, which certainly improve the quality and comprehensibility of our work.
All suggested corrections have been made in the manuscript:
two cartoons representing CCR4 role in tumorigenesis and in Mogalizumab therapy have been added,
Introduction has been expanded,
References of various cancer cells expressing CCR4 have been added,
CCR4 in animal studies has been added,
References regarding CCR4-receptor signaling have been added,
Conclusion has been revised
Round 2
Reviewer 3 Report
The authors have answered all of my questions, and the paper has significantly improved.